# Presence of Pain Shows Greater Effect than Tendon Structural Alignment During Landing Dynamics

**DOI:** 10.3390/jfmk10010074

**Published:** 2025-02-24

**Authors:** Silvia Ortega-Cebrián, Diogo C. F. Silva, Daniela F. Carneiro, Victor Zárate, Leonel A. T. Alves, Diana C. Guedes, Carlos A. Zárate-Tejero, Aïda Cadellans-Arróniz, António Mesquita Montes

**Affiliations:** 1Physiotherapy Department, Facultat Fisioteràpia, Universitat Internacional de Catalunya (UIC), Sant Cugat de Vallès, 08017 Barcelona, Spain; vzarate@uic.es (V.Z.); czarate@uic.es (C.A.Z.-T.); acadellans@uic.es (A.C.-A.); 2Physiotherapy Department, Santa Maria Health School (ESSSM), 4049-024 Porto, Portugal; diogo.silva@santamariasaude.pt; 3CIR, E2S, Polytechnic of Porto, 4200-072 Porto, Portugal; danielaferreiracarneiro@gmail.com (D.F.C.); leonel00alves00@gmail.com (L.A.T.A.); dianaguedes75@gmail.com (D.C.G.); antoniomesquitamontes@gmail.com (A.M.M.); 4Functional Sciences Department, School of Health of Polytechnic of Porto (ESS—Porto), 4200-070 Porto, Portugal; 5Physiotherapy Department, E2S, Polytechnic of Porto, 4200-072 Porto, Portugal

**Keywords:** volleyball, patellar tendon, athletic injuries, imaging, biomechanics

## Abstract

**Background/Objectives**: Eccentric loading during landing is considered a key factor in the development of patellar tendinopathy and is associated with stiff landings and patellar tendinopathy. This study aims to investigate the relationship between tendon structure, presence of pain, and sex differences in landing kinematics and kinetics during countermovement jumps (CMJ) and tuck jump tests (TJT) in professional volleyball players. **Methods**: Professional volleyball players aged 18 to 30 years old (14 females and 25 males) participated in a cross-sectional study. Data included the Victorian Institute of Sport Assessment Patellar Tendon (VISA-P) questionnaire; patellar tendon ultrasound characterization tissue (UTC) scans, in order to identify groups with misaligned tendon fibers (MTF) or aligned tendon fibers (ATF); and biomechanical assessments for CMJ and TJT. The joint angle (JA) at the lower limb was measured at peak ground reaction force (peak_vGRF) and maximal knee flexion (max_KF). A general linear model was used to evaluate joint JA differences between tendon alignment, pain, and sex variables. Sample *t*-tests compared peak_vGRF, load time, load rate, and area based on tendon alignment, pain presence, sex, and jump. The statistical significance of *p*-value is >0.05, and the effect size (ES) was also calculated. **Results**: The MTF group revealed decreased knee JA during TJT at peak_vGRF (*p* = 0.01; ES = −0.66) and max_KF (*p* = 0.02; ES = −0.23). The presence of pain was associated with increased JA during the CMJ, particularly at peak_vGRF and max_KF for trunk, hip, and ankle joints. Females showed decreased peak_vGRF than males. Landing with misaligned tendon fibers showed longer load times compared to aligned tendon fibers (*p* = 0.021; ES = −0.80). The TJT exhibited a greater load rate than the CMJ (*p* = 0.00; ES = −0.62). **Conclusions**: Pain is a critical factor influencing greater JA during landing, particularly at the trunk, hip, and ankle joints in CMJ. Misaligned tendon fibers compromise landing dynamics by increasing trunk JA during TJT. Kinetics varied significantly by sex and jump type, while pain and tendon structure revealed limited differences.

## 1. Introduction

Patellar tendinopathy is a common overuse injury mainly characterized by pain, reduced strength, and functional loss [1]. Histopathological changes frequently occur in the tendon matrix, resulting in structural disorganization and diminished capacity to withstand mechanical loads [2]. Pathophysiology conditions are complex and multifactorial, with diagnosis typically based on clinical symptoms such as pain, reduced strength, and function, alongside structural changes observed in imaging studies. However, structural alterations in the tendon are not always pathological, as tendon structural adaptations without associated pain have been reported [3,4]. Despite that, imaging-based assessments of tendon structure, when combined with pain evaluation, remain a cornerstone of patellar tendinopathy diagnosis [5]. This highlights the challenges in determining the threshold of structural changes necessary for symptom manifestation [6].

Theories regarding patellar tendinopathy suggest it is a progressive condition initiated by acute excessive loading, leading to acute reactive tendinopathy [7,8,9]. This phase is mediated by nociceptive agents, including acetylcholine, glutamine, substance P, and catecholamines [8,9,10]. If overloading persists, structural abnormalities within the tendon matrix can develop, exceeding the tendon’s apoptotic regulation capacity [11,12]. Pain symptoms may arise as a protective mechanism to prevent further damage, potentially linked to altered pain modulation and sensitization processes [3,13]. Chronic overloading can result in tendon degeneration, marked by collagen fiber disruption and matrix disorganization [14].

Tendon injuries are prevalent in sports involving repetitive jumping, such as volleyball and basketball, where knee extension during take-off and landing imposes significant mechanical stress on the tendon [15]. Male athletes in these sports demonstrate a particularly high incidence of patellar tendinopathy [15,16], though it is also observed in female athletes participating in volleyball, basketball, and soccer [15,17,18]. Risk factors include the high mechanical demands of vertical jumping, as well as acceleration, deceleration, and directional changes [19]. Recent studies have begun distinguishing the effects of horizontal versus vertical loading on the tendon [20]. Vertical loading, often related in the context of “Jumper’s Knee”, described by Dr. Martin Blazina in 1973, is associated with alterations in the proximal tendon matrix, leading to chronic morbidity and disability [21,22,23].

Patellar tendinopathy is closely linked to the rate and magnitude of tendon loading [20]. While most biomechanical studies have focused on initial contact (IC) parameters, including joint angles, angular velocities, and ground reaction forces, these factors show limited correlation with patellar tendinopathy [24]. A significant research gap exists in examining landing phases after IC, particularly during the eccentric loading phase, which spans from touchdown to the lowest center of mass position [25]. Eccentric loading during landing is considered a key factor in patellar tendinopathy development, as well as being related to stiff landing [26].

Stiff landing patterns, characterized by a restricted post-touchdown range of motion and shorter landing durations, have been linked to the onset of patellar tendinopathy [27]. Moderate evidence showed in meta-analyses associated reduced ankle dorsiflexion and decreased knee joint power during volleyball approaches and drop landings with patellar tendinopathy [24]. Structural abnormalities such as reduced tenocyte adhesion and impaired collagen organization may diminish the tendon’s energy absorption capacity, particularly during eccentric phases [2,24,28].

Structural changes in the patellar tendon, often identified via hypoechoic imaging through standard ultrasound (US), include increased proteoglycan and water content, which create inter-tenocyte spacing [28]. The amount of hypoechoic imaging is related to the amount of unstructured tendon fibers, although the US is a controversial method to measure the amount of tendon structural changes [29]. Ultrasound Tissue Characterization (UTC) has become a valuable tool for quantifying tendon structure and evaluating loading or treatment effects, particularly in the patellar and Achilles tendons [30]. UTC classifies tendon quality into four echo types, with echo types I and II indicating healthier tendon structure and echo types III and IV reflecting degeneration when exceeding 10% [31]. While UTC is not a diagnostic tool, it provides insights into structural changes in symptomatic and asymptomatic individuals’ tendons [19,32].

Clinical assessments of patellar tendinopathy frequently identify stiff knee landing patterns involving limited joint range of motion and increased knee extensor activity during eccentric phases. Although drop jump and countermovement jump (CMJ) tests are widely studied, there is limited research on repetitive high-intensity demands tests such as tuck jump test (TJT), which impose greater eccentric loads [33]. Biomechanical changes during landing in professional volleyball players, a population with a high prevalence of patellar tendinopathy, have been studied, although landing dynamic changes and tendon structure adaptations remain unexplored [23]. Additionally, studies on sex-based differences in landing dynamics are scarce in the literature.

This study aimed to compare trunk, hip, knee, and ankle joint angles (JA) during the eccentric landing phase between players with aligned and misaligned tendon fibers, with pain and no pain, and between female and male professional volleyball players at maximal knee flexion (max_KF) and, the peak of vertical ground reaction force (peak_vGRF). In addition, peak_vGRF, time, load rate, and area were compared between aligned and misaligned tendon fibers, with pain and no pain, and between female and male professional volleyball players and CMJ and TJT.

The findings may provide insights into whether pain, rather than structural changes alone, drives mechanical adaptations during landing. Moreover, the study could elucidate sex-based differences in landing dynamics among professional volleyball players.

## 2. Materials and Methods

### 2.1. Study Design

A cross-sectional study, including 39 professional female (*n* = 22) and male (*n* = 17) volleyball players, was conducted during the postseason in July 2023. Participants were recruited through verbal information from physical coaches and university staff affiliated with the Portuguese Volleyball Federation and Polytechnic Institute of Porto. Reporting follows Strengthening The Reporting of Observational Studies in Epidemiology (STROBE) Guidelines [34]. Data collection was conducted at the biomechanical laboratory of the Centre for Research and Rehabilitation at the Polytechnic Institute of Porto. The study adhered to the Declaration of Helsinki and received ethical approval from the Research Ethics Committee at Universitat Internacional de Catalunya (FIS-2022-06). The study was registered in clinicaltial.gov (NCT06829056). Before any intervention, participants were briefed on the study protocol and provided informed consent prior to data collection.

### 2.2. Inclusion/Exclusion Criteria

Inclusion criteria included volunteer participants aged between 18–30 years old, competing in the highest-level Portuguese volleyball league with a minimum of 10 years of volleyball practice. Those participants in the presence of injuries or discomfort that impeded maximal jumping tests, history of surgery within the last 12 months, or illness during data collection were excluded from the study.

### 2.3. Outcome Measures

The primary outcome was the joint angle (JA) of the trunk, hip, knee, and ankle between aligned tendon fibers (AFT) and misaligned tendon fibers (MTF), measured during peak vertical ground reaction force (peak_vGRF) and maximal knee flexion (max_KF). Secondary outcomes included the presence of pain (yes/no) and sex (female/male). Further secondary outcomes were kinematic data of peak_vGRF, load time and rate, and knee area between tendon alignment, pain, sex, and type of jump.

Landing kinematics were calculated as peak_vGRF, which describes the peak force during landing, and the load time of the knee is calculated by the duration from initial contact to max_KF, representing the eccentric landing phase. The load rate of the knee considers the vertical ground reaction force (vGRF) at maximal knee flexion divided by the load time of the knee, describing the rate at which the ground reaction force increases as the knee flexes, representing how quickly the knee joint adapts to the applied forces. Lastly, the area is described by the area under the curve between IC and max_KF, reflecting the total impulse area during this phase, indicating the accumulated force applied over time [33,35,36]. All the outcomes were recorded in one session.

### 2.4. Procedure

Data collection comprised three steps:

1. Questionnaires: Participants completed a demographic (age, height, and weight) and pain-related questionnaire, including the Victorian Institute of Sport Assessment Patellar Tendon (VISA-P), to quantify pain symptoms during volleyball practice. Participants with VISA-P scores > 50 were categorized as having pain.

2. Bilateral patellar tendon UTC scans. UTC scans were performed using ultrasound tissue characterization (UTC). Participants were seated with knees flexed at 90°. All tests were conducted by a single tester (SOC), who performed all UTC scans to decrease possible inter-tester biases [37]. UTC scans were performed using B-mode ultrasound with a linear transducer of 7–10 MHz (SmartProbe 10L5; Terason 2000, Teratech, Rockville, MD, USA). An ultrasound probe (SmartProbe 12L5-V, Terason 2000+; Teratech) was fixed to a tracking device (UTC Tracker, UTC Imaging, Stein, The Netherlands) that automatically moved the transducer on the perpendicular axis of the tendon and recorded cross images at 0.2 mm intervals [38]. Window size 17 was used for imaging analysis, and the region of interest (ROI) was located around the tendon in the transverse view, with contours of the ROI drawn at the proximal tendon (10 and 20% tendon length) and mid-tendon (50% tendon length). Total tendon length was determined from the inferior angle of the patella to the most proximal tibia tuberosity. Tendon contours were marked by an experienced investigator (DF), increasing the reliability of UTC outcomes [32]. Echo types were quantified through the UTC Imagin System 2.0 (UTC 2010) [30]. Tendon structure was classified into four echo types (I–IV) based on consistency and hypoechoic region: echo types I and II represented healthy (I) and adapted (II) aligned fibrillar structure, and echo types III and IV represented structural abnormalities (III) and degenerative tendon structure (IV). Group assignment criteria were defined as aligned fibrillar tendon (AFT) for echo types I and II, being >90%; and misaligned fibrillar tendon (MFT) for echo types III and IV, being >10%.

3. 3D Motion Capture of Jumping Tests. Before the jumping tests, participants performed a 5-min warm-up on a static bicycle, achieving 125 watts during the final 2 min. During a 5-min rest, retroreflective markers were attached to anatomical landmarks. Bony landmarks included acromion, fifth lumbar spine, posterior superior iliac spine, anterior superior iliac spine, lateral aspect of the thighs, lateral knee epicondyle, peroneal malleolus, and apophysis of the fifth metatarsal in accordance with the biomechanical model used in the Plug-in2 Rosen et Gait module of the data collection software 3.1 [38,39]. Kinematic data were obtained by an optoelectronic system, Qualisys Motion Capture System (Qualisys AB, Göteborg, Sweden) was used for motion capture, 12 infrared cameras, 8 Oqus 500 and 4 Miqus M3, co, at a sampling frequency of 100 Hz. The image capture hardware was connected to the Qualisys USB Analog Acquisition interface to synchronize the kinetic and kinematic data with the Qualisys Track Manager (QTM software version 2022.1 (Kvarnbergsgatan 2, Göteborg, Sweden). Participants performed two repetitions of maximal CMJ and TJT. CMJ required a bilateral jump with hands on the waist aiming for maximal height, while TJT involved three continuous “knees-to-chest” high-intensity jumps. Familiarization trials were omitted as participants were already familiar with the tests. The vertical ground reaction force was collected using a force platform (model FP4060-08; Bertec Corporation, Columbus, OH, USA) connected to a signal amplifier (model AM6300; Bertec Corporation) with a frequency of 100 Hz.

### 2.5. Sample Size Calculation

The sample size was calculated using the GRANMO calculator based on previous studies, with echo type II as the primary outcome [40]. Using an alpha risk of 0.05, a beta risk of 0.15, and an anticipated dropout rate, a sample size of 19 participants per group was determined to ensure sufficient statistical power.

### 2.6. Statistical Analysis

Descriptive statistics were calculated, and normality was assessed using the Shapiro–Wilk test, which indicated a normal distribution. Data are presented as means and standard deviations (SD). A Levene’s test for homogeneity of variances was conducted to compare quantitative data between the aligned fibrillar tendon (AFT) and misaligned fibrillar tendon (MFT) groups. For UTC tendon analysis, the intraclass correlation coefficient (ICC) was calculated based on data from previous studies that used the same methods for the proximal and mid-tendon regions [30]. These studies reported high reproducibility and moderate-to-excellent intra-observer reliability for UTC assessments in trained testers (ICC > 0.80) [40]. A general linear model (GLM) was used to evaluate joint angle (JA) at the trunk, hip, knee, and ankle between tendon alignment with Bonferroni correction applied (*p* < 0.005). Potential interactions between factors such as pain and sex were also assessed using a multivariate GLM. This method allowed for the consideration of both fixed and random effects, providing a robust analytical framework. Additionally, 95% confidence intervals (CI), *p*-values, and effect sizes were reported for all parameters. Effect size magnitudes (Cohen’s d) were interpreted as follows: ≤0.19 = trivial, 0.20–0.49 = small, 0.50–0.79 = moderate, and ≥0.80 = large (7). Moderate or large effect sizes were considered substantial [40,41]. Independent sample *t*-tests were used to compare variables such as peak vGRF, load time, load rate, and knee area between tendon alignment, presence of pain, sex, and type of jump. The significance threshold was set at an alpha level of 0.05. All statistical analyses were conducted using SPSS version 26 (IBM, Amarok, NY, USA).

## 3. Results

After data processing, 15 male patellar tendons were excluded from the analysis due to poor tendon imaging or inaccurate kinematics recordings (See Figure 1). For data analysis, each patellar tendon (*n* = 62) was considered and classified into the aligned fibrillar tendon group (AFT) (*n* = 38) or the misaligned fibrillar tendon group (MFT) (*n* = 24). Demographic data are presented in Table 1.

Demographic data showed that aligned and misaligned fibrillar tendon groups were significantly different in age and VISA-P. The AFT group consisted of younger players compared to the MFT group. However, they exhibited higher levels of dysfunction due to the presence of pain, as indicated by a VISA-P score of 54, compared to 74 for the MFT group (*p* < 0.005).

Descriptive data of mean, standard deviation, and 95% confidence interval (95% inferior and superior limit) for each dependent and independent variable for each group are presented in Table 2, as well as *p*-values and effect size. Figure 1 shows significant differences in JA at the trunk, hip, knee, and ankle between tendon structure, pain, and sex.

### 3.1. Joint Kinematic Data

During the CMJ, tendon fibrillar alignment did not show any differences in JA at any of the joints assessed, with a trivial effect between groups either in peak_vGRF or max_KF. Differences were found during the tuck jump test at the knee joint in both peak_vGRF (*p* = 0.012; ES = −0.66) and max_KF (*p* = 0.028; ES = −0.23). Results showed less degrees of knee flexion in the MFT group than (129.73° at peak_vGRF; 117.03° at max_KF) in the AFT group (126.48° at peak_vGRF; 115.22° at max_KF) with a moderate effect size for the peak_vGRF and small for the maximal KF.

Figure 2 shows the statistically significant differences of JA at the trunk, hip, knee, and ankle between tendon structure, pain, and sex.

In this study, the presence of pain showed increased JA at peak_vGRF and max_KF at the trunk, hip, and ankle joints during CMJ landing kinematics. For peak_vGRF; trunk (*p* = 0.037; ES = 0.12), hip (*p* = 0.031; ES = 0.56), and ankle (*p* < 0.01; ES = 0.37). For max_KF and trunk (*p* = 0.039; ES = 0.09, hip (*p* = 0.023; ES = 0.14), and ankle (*p* = 0.027; ES = 0.45). The presence of pain resulted in greater JA at the trunk, hip, and ankle joints than pain-free tendons. During TJT, JA in the presence of pain JA was only greater at the trunk for peak_vGRF (*p* = 0.041; ES = 0.14) and max_KF (*p* = 0.029; ES = 0.15).

Differences in JA between female and male professional volleyball players were not found during CMJ, either in peak_vGRF or max_KF. During the tuck jump test, male volleyball players exhibited lesser JA at the trunk compared to female players in both peak_vGRF (*p* = 0.024; ES = 0.02) and max_KF (*p* < 0.01; ES = 0.03). Female players appeared to land with more trunk flexion than their male counterparts.

### 3.2. Join Kinetic Data

Tendon structure and presence of pain did not seem to show significant changes in peak_vGRF during landing, only sex (*p* = 0.011; ES = −0.48) and type of jumps (*p* < 0.01; ES = −0.8). Male professional volleyball players exhibited greater peak_vGRF than females (2651.36 N ± 1240.18 N and 2307.37 ± 838.78, respectively) with a moderate effect size of −0.48. In addition, TJT showed nearly 50% more landing reaction force than CMJ (1811.85 N ± 801.59 N and 3117.4 N ± 853.11 N, respectively).

Regarding load time during landing, MFT (0.2 s ± 0.17 s) demonstrated a longer load time than AFT (0.18 s ± 0.03 s) with a large effect size (*p* = 0.028; ES = −0.80). Results did not show differences in load time in the presence of pain, sex, or type of jumps.

Load rate showed a significant difference only between CMJ and TJT, whereas TJT (20,021.33 ± 10,539.85) showed nearly three times greater load rate than CMJ (7424.34 ± 2232.83) with a moderate effect size (*p* < 0.01; ES = −0.62).

The results of the difference of area at the knee joint showed that males (389.24 ± 118.43) demonstrated greater knee joint area than females (290.42 ± 85.21; *p* = 0.00; ES = −0.73) and TJT (357.88 ± 146.39) greater than CMJ (313.3 ± 56.16; (*p* < 0.01; ES = −0.86). Table 3 presents descriptive data of mean, standard deviation, 95% confidence interval (superior and inferior limit), *p*-value, and effect size of kinetics data.

## 4. Discussion

Landing dynamics have previously been associated with patellar tendinopathy and patellar pain [42], but the evidence is limited in their relationship with patellar tendon structural abnormalities, and it remains unclear. To our knowledge, this is the first study to investigate landing dynamics and patellar tendon structure during two distinct types of bilateral vertical jumps: the countermovement jump (CMJ) and the repetitive high-intense tuck jump test (TJT). The goal of comparing these jumps was to evaluate differences in landing dynamics between aligned fibrillar tendons (normal patellar tendon structure) and misaligned fibrillar tendons (indicative of structural abnormalities). We hypothesized that landing dynamics would differ between CMJ and TJT due to reduced energy storage capacity in tendons with altered cellular matrix [43].

We could confirm our hypothesis only for the TJT as the MTF group showed decreased knee JA at peak_vGRF and max_KF. The main factor affecting JA during the jumping tests was the presence of pain. The presence of pain was associated with increased JA during the CMJ, particularly at peak_vGRF and max_KF for trunk, hip, and ankle joints. No sex differences were observed in JA during the CMJ, although females showed decreased peak_vGRF than males. Regarding kinetics, MFT showed a longer load time during the eccentric phase of landing compared to AFT. The TJT exhibited a greater load rate than the CMJ. Male players displayed greater knee areas during the TJT compared to female players.

Demographic data revealed that older volleyball players more frequently exhibited misaligned fibrillar tendons (MFT), which corresponded with greater years of volleyball practice. Conversely, 10% of younger players with aligned fibrillar tendons (AFT) and fewer years of experience reported greater pain, as reflected by VISA-P scores. Although decreased pain sensitivity is age-related due to somatosensory inhibition [11], in our sample, there were only 2–5 years of difference between groups. We suggest that tendon pain is a nociceptive response non-related to tissue damage [44]. Perhaps younger volleyball players report greater levels of pain because they are less affected by the differential effects of sensory discrimination and affective emotional components of playing pain-free, whereas in older players, the cultural sport belief of playing with tendon pain is commonly accepted [13,44].

Our findings in kinematics of landing dynamics showed no significant difference in patellar tendon structure on the joint angle during CMJ landings. These results align with previous studies [24,33], which reported no kinematic differences at the hip, knee, and ankle between athletes with asymptomatic patellar tendon abnormalities and control groups. Similarly, Scattone (2016) found no significant JA differences between groups with or without patellar tendon abnormalities [43].

For the TJT, however, JA at the knee differed between groups. Players with MFT exhibited smaller JA at peak_vGRF and max_KF. These findings corroborate other studies that have identified reduced knee flexion as a strong predictor of patellar tendinopathy and tendon abnormalities [45]. Further altered knee joint angles, such as increased abduction and internal rotation during landing, have been implicated in increased loading on the proximal patellar tendon [46,47]. These results suggest that tendon structural changes may influence landing dynamics during high demand in terms of great ground reaction force, greater load time, and motor control, such as the TJT, but not during single vertical jumps like the CMJ.

The presence of pain significantly affected landing kinematics. Players with pain exhibited greater JA at the trunk, hip, and ankle during peak_vGRF and max_KF for CMJ landings. Similar findings were reported by Scattone (2016), who observed increased trunk and ankle JA in symptomatic patellar tendinopathy groups [43]. Greater trunk flexion during landing has been suggested as a compensatory mechanism to reduce patellar tendon load and alleviate pain [43]. Although there are no biomechanical explanations, this interpretation is supported by our results, which demonstrated greater trunk JA in symptomatic players even during explosive jumps like the TJT.

Regarding hip JA in the presence of pain remains less understood. Previous studies have shown inconsistent results [23,24,43]. Our findings indicate greater hip JA during landing in painful tendons. Knee JA, however, showed no significant differences in the presence of pain during either CMJ or TJT landings, with similar values observed at peak_vGRF and max_KF across all conditions. For the ankle, our results diverge from prior studies, which often report reduced dorsiflexion in symptomatic tendons [24]. Instead, we observed greater dorsiflexion in the symptomatic group, aligning with findings from Scattone (2016) [43]. We could extrapolate results to suggest that greater trunk JA could occur as a compensatory mechanism to reduce patellar tendon pain and load, as decreased peak patellar tendon load, knee extensor moment, and pain with flex trunk position during landing have been reported before [43].

Sex-based differences in landing dynamics were observed only in trunk JA during TJT landings, where female players exhibited less trunk flexion compared to males. Female players tended to land with less trunk, hip, and ankle flexion but more knee flexion compared to males, although these differences were not statistically significant except for trunk JA during the TJT. Further research between the sexes is needed because most of the studies have been performed on male players or with small female sample sizes [24].

Regarding the kinetics of landing dynamics, neither tendon structure (MFT vs. AFT) nor pain significantly affected peak_vGRF. This finding supports prior evidence suggesting no direct relationship between pain and peak_vGRF in landing dynamics. Interestingly, our results demonstrated for the first time that tendon structure itself does not influence peak_vGRF. These findings suggest that other musculoskeletal interfaces, such as the hip or ankle, may compensate to maintain functional landing mechanics despite tendon abnormalities [25,47].

Expectedly, sex and jump type influenced peak_vGRF. Male players generated higher peak_vGRF than females, and the TJT elicited nearly 60% higher peak_vGRF than the CMJ. This highlights the importance of considering jump type and sex in biomechanical assessments of landing.

For load time, only MFT has been shown to be slower in the eccentric landing phases than AFT. Our results support the limited evidence suggesting that stiff landings may reduce load time [48]. Previous evidence has associated increased load time with stiff landing and patellar tendinopathy while increasing load rate [24]. Even though we found slower load time between tendon alignment groups, load rate showed no significant differences across tendon structure, pain, or sex. Predictably, the repetitive and high-intensity nature of the TJT resulted in higher load rates than the CMJ. As well as higher knee area between sex and type of jump, corresponding to a greater physical difference between sex and greater physical demands between the two types of jumps.

### 4.1. Clinical Implications

Our findings suggest that MFT tendons could maintain energy absorption capacity and optimal landing dynamics during single vertical jumps (CMJ) but may not perform similarly under repetitive, high-demand conditions (TJT). Pain appears to be a primary factor influencing JA changes during landing, particularly at the trunk, hip, and ankle, rather than tendon structure alone. These results raise the question of whether structural changes in the tendon are causative of injury or adaptive responses to loading demands and pain. Our results challenge the reliance on imaging alone for diagnosing patellar tendinopathy. While tendon abnormalities may indicate injury risk, they do not necessarily affect kinematics in standard jump tests like the CMJ. However, they may influence landing mechanics under high-load conditions. Clinicians should focus on testing athletes under high-energy storage demands and correcting trunk posture to potentially reduce patellar tendon load. Additionally, maintaining hip and ankle JA during landing may help mitigate injury risk.

### 4.2. Limitations

This study’s cross-sectional design and the small number of participants in the MFT group limits the ability to establish causality between landing dynamics and tendon structural changes. A prospective study is warranted to investigate whether altered landing dynamics contribute to tendon adaptations or injuries. Finally, while the TJT imposed higher biomechanical demands than the CMJ, incorporating additional sports-specific tasks may provide greater ecological validity.

## 5. Conclusions

MFT tendons preserve energy absorption capacity and landing dynamics during CMJ but not during repetitive, high-demand jumps like the TJT. The presence of pain is a key factor influencing JA differences during landing, particularly increasing at the trunk, hip, and ankle. Male volleyball players showed greater JA at the trunk compared to female players. Peak_vGRF, load time, rate, and knee area were greater for TJT than CMJ. Male players showed greater peak_vGRF and knee area than females. MTF showed longer load times than AFT. These findings suggest that tendon structural changes may represent adaptive responses rather than direct injury mechanisms. Future research should explore the role of landing dynamics in tendon adaptations and injury prevention strategies.

## Figures and Tables

**Figure 1 jfmk-10-00074-f001:**
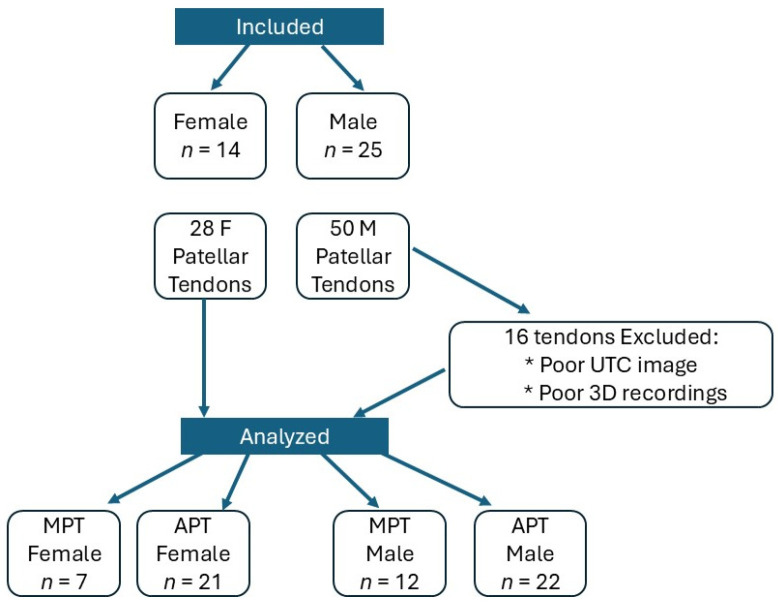
Group definition flowchart. UTC = ultrasound tissue characterization; MFT = misaligned fibrillar tendon.

**Figure 2 jfmk-10-00074-f002:**
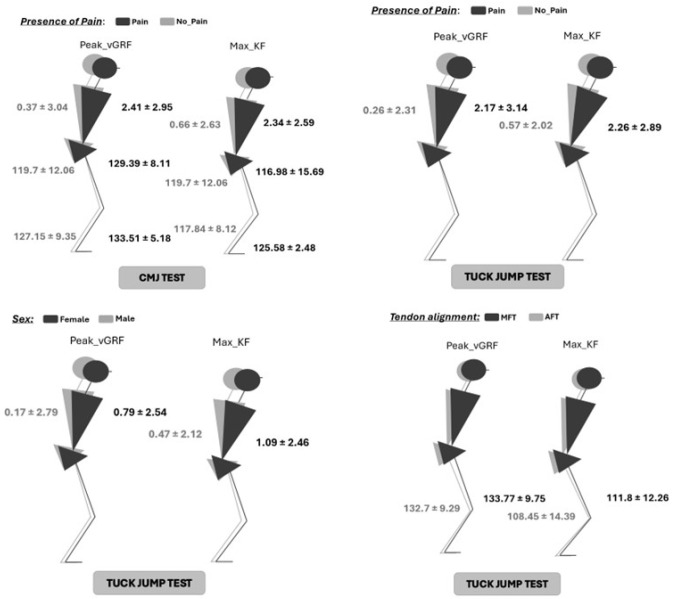
Significant differences of JA at the trunk, hip, knee, and ankle between tendon structure, pain, and sex. *p*-value < 0.005; AFT = aligned fibrillar tendon; MFT = misaligned fibrillar tendon; CMJ = counter movement jump; Peak vGRF = peak vertical ground reaction force; Max_KF = maximal knee flexion.

**Table 1 jfmk-10-00074-t001:** Demographic data.

	Mean	SD
AFT (female *n* = 21; male *n* = 28)		
Age *	24.95	3.16
Height	175.66	13.92
Weight	89.74	8.52
Years of practice	13.52	3.67
Pain *n* = 5		
VISA-P *	54	7.85
MFT (female *n* = 7; male *n* = 12)		
Age	26.32	2.98
Height	179.86	11.71
Weight	86.49	9.14
Years of practice	14.78	4.12
Pain *n* = 12		
VISA-P	76	12.37

* = *p* < 0.005. Data are presented in mean ± standard deviation; abbreviations: VISA-P (Victorian Institute Sports Assessment Patellar).

**Table 2 jfmk-10-00074-t002:** Descriptive statistics, comparison of trunk, hip, knee, and ankle range of motion (degrees) between tendon alignment, presence of pain, and sex in professional volleyball players during CMJ and TJT.

Factor	Moment	Join	Variable	*n*	Mean ± SD	95% CI (Inf to Sup Limit)	*p*-Value	Effect Size	Mean ± SD	95% CI (Inf to Sup Limit)	*p*-Value	Effect Size
Tendon	*Peak_vGRF*	Trunk	AFT	38	0.48 ± 3.42	−0.42 to 1.38	0.52	−0.19	0.34 ± 2.88	−0.4 to 1.08	0.334	0.04
			MFT	24	1.19 ± 1.94	0.07 to 2.31			1.23 ± 1.21	0.53 to 1.92		
		Hip	AFT	38	121.89 ± 12.67	118.56 to 125.23	0.80	0.05	124.55 ± 12.03	121.44 to 127.66	0.207	−0.73
			MFT	24	119.77 ± 8.31	114.97 to 124.57			117.96 ± 8.69	112.94 to 122.97		
		Knee	AFT	38	133.77 ± 9.75	131.21 to 136.34	0.60	0.02	129.73 ± 6.69	128 to 131.46	0.012 *	−0.66
			MFT	24	132.7 ± 9.29	127.34 to 138.06			126.48 ± 5.94	123.05 to 129.91		
		Ankle	AFT	38	128.65 ± 8.86	126.32 to 130.98	0.32	0.06	125.67 ± 10.36	122.99 to 128.34	0.956	−0.26
			MFT	24	126.14 ± 9.75	120.51 to 131.77			123.71 ± 8.32	118.91 to 128.52		
	*Max_KF*	Trunk	AFT	38	0.71 ± 2.8	−0.02 to 1.45	0.35	−0.02	0.74 ± 2.49	0.1 to 1.38	0.573	0.02
			MFT	24	1.55 ± 2.3	0.22 to 2.88			1.09 ± 1.31	0.33 to 1.84		
		Hip	AFT	38	103.98 ± 23.94	97.68 to 110.27	0.33	0.23	117.6 ± 15.14	113.69 to 121.51	0.166	−0.63
			MFT	24	95.12 ± 27.48	79.25 to 110.98			110.38 ± 9.2	105.07 to 115.69		
		Knee	AFT	38	111.8 ± 12.26	108.57 to 115.02	0.367	0.08	117.03 ± 9.85	114.49 to 119.58	0.028 *	−0.23
			MFT	24	108.45 ± 14.39	100.14 to 116.75			115.22 ± 10.71	109.04 to 121.4		
		Ankle	AFT	38	119.79 ± 7.76	117.75 to 121.83	0.14	0.09	119.62 ± 11	116.77 to 122.46	0.985	−0.16
			MFT	24	116.28 ± 8.44	121.16 to 121.16			118.24 ± 9.69	112.65 to 123.84		
Pain	*Peak_vGRF*	Trunk	P	17	2.41 ± 2.95	0.54 to 4.29	0.037 *	0.12	2.17 ± 3.14	0.28 to 4.07	0.041 *	0.14
			N_P	45	0.37 ± 3.04	−0.42 to 1.16			0.26 ± 2.31	-0.34 to 0.85		
		Hip	P	17	129.39 ± 8.11	124.24 to 134.55	0.031 *	0.56	128.53 ± 6.62	124.53 to 132.53	0.191	1.02
			N_P	45	119.79 ± 12.06	116.65 to 122.93			122.22 ± 12.39	119.02 to 125.42		
		Knee	P	17	133.92 ± 10.18	127.45 to 140.38	0.38	0.01	129.04 ± 7.44	124.54 to 133.53	0.186	−0.04
			N_P	45	133.64 ± 9.6	131.13 to 136.14			129.19 ± 6.57	127.5 to 130.89		
		Ankle	P	17	133.51 ± 5.18	130.22 to 136.8	0.00 *	0.37	133.92 ± 7.66	129.29 to 138.55	0.061	2.20
			N_P	45	127.15 ± 9.35	124.71 to 129.59			123.28 ± 9.46	120.84 to 125.73		
	*Max_KF*	Trunk	P	17	2.34 ± 2.59	0.7 to 3.99	0.039*	0.09	2.26 ± 2.86	0.54 to 3.99	0.029 *	0.15
			N_P	45	0.66 ± 2.63	−0.03 to 1.34			0.57 ± 2.02	0.05 to 1.09		
		Hip	P	17	116.98 ± 15.69	107.01 to 126.95	0.023 *	0.14	123.49 ± 10.55	117.11 to 129.86	0.254	1.14
			N_P	45	98.89 ± 25.23	92.32 to 105.47			114.62 ± 14.86	110.78 to 118.46		
		Knee	P	17	113.9 ± 14.43	104.73 to 123.07	0.83	0.19	117.92 ± 13.22	109.93 to 125.9	0.157	0.27
			N_P	45	110.53 ± 12.44	107.29 to 113.77			116.42 ± 9.33	114.01 to 118.83		
		Ankle	P	17	125.58 ± 2.48	124.01 to 127.16	0.027 *	0.45	128.39 ± 8.14	123.48 to 133.31	0.075	2.10
			N_P	45	117.84 ± 8.12	115.73 to 119.96			117.2 ± 10.18	114.57 to 119.83		
Sex	*Peak_vGRF*	Trunk	F	28	0.76 ± 3.13	−0.25 to 1.78	0.60	0.01	0.79 ± 2.54	−0.02 to 1.6	0.024 *	0.02
			M	34	0.45 ± 3.29	−0.72 to 1.62			0.17 ± 2.79	−0.8 to 1.15		
		Hip	F	28	118.76 ± 10.78	115.27 to 122.25	0.47	−0.12	121.61 ± 10.41	118.28 to 124.94	0.458	−0.26
			M	34	124.7 ± 12.57	120.24 to 129.15			125.3 ± 12.95	120.78 to 129.82		
		Knee	F	28	134.28 ± 9.27	131.27 to 137.28	0.59	0.05	129.93 ± 5.96	128.03 to 131.84	0.358	0.14
			M	34	132.72 ± 10.08	129.15 to 136.29			128.16 ± 7.34	125.6 to 130.72		
		Ankle	F	28	128.28 ± 9.08	125.33 to 131.22	0.99	0.00	124.27 ± 10.49	120.91 to 127.62	0.449	−0.17
			M	34	128.02 ± 9.1	124.79 to 131.25			126.51 ± 9.37	123.24 to 129.78		
	*Max_KF*	Trunk	F	28	1.02 ± 2.74	0.13 to 1.91	0.64	0.011	1.09 ± 2.46	0.31 to 1.88	0.000 *	0.03
			M	34	0.7 ± 2.72	−0.26 to 1.67			0.47 ± 2.12	−0.27 to 1.21		
		Hip	F	28	98.51 ± 23.25	90.97 to 106.05	0.11	−0.29	112.91 ± 13.11	108.71 to 117.1	0.744	−0.46
			M	34	106.68 ± 26	97.46 to 115.9			120.15 ± 15.12	114.87 to 125.43		
		Knee	F	28	111.4 ± 12.63	107.31 to 115.5	0.50	0.02	116.87 ± 9.43	113.85 to 119.89	0.614	0.03
			M	34	110.84 ± 12.89	106.27 to 115.41			116.48 ± 10.71	112.74 to 120.21		
		Ankle	F	28	118.44 ± 7.53	116 to 120.88	0.51	−0.05	117.62 ± 11.07	114.08 to 121.16	0.456	−0.28
			M	34	119.9 ± 8.48	116.89 to 122.9			121.4 ± 10.06	117.89 to 124.91		

* = *p* < 0.005; CI = coeficient interval; GRF = ground reaction force; AFT = aligned fibrillar tendon; MFT = misaligned fibrillar tendon; P = pain; N_P = no pain; F = female; M = male; CMJ = counter movement jump; TJT = tuck jump test.

**Table 3 jfmk-10-00074-t003:** Comparison of peak_vGRF, time and load rate, and knee area between tendon alignment, presence of pain, sex, and type of jump in male professional volleyball players.

Factor	Variable	Tendon Alingment	*n*	Mean ± SD	95% CI (Sup to Inf)	*p*-Value	Effect Size
Peak_vGRF (N)	*Tendon Structure*	AFT	208	2.44 ± 10.49	26.93 to 21.99	0.871	−0.56
		MFT	72	2.47 ± 10.52	26.15 to 23.26		
	*Pain*	No_Pain	112	24.75 ± 10.61	26. 32 to 23. 18	0.836	0.84
		Pain	110	2.44 ± 10.82	26. 51 to 22. 42		
	*Sex*	Female	152	23.07± 8.38	24.41 to 21.72	0.011 *	−0.48
		Male	128	26.51 ± 12.40	2868.27 to 2434.45		
	*Jump*	CMJ	140	18.11 ± 8.01	19.45 to 16.7	0.000 *	−0.02
		TJT	140	31.17 ± 8.53	32.59 to 29.74		
Load_Time/Knee (sec)	*Tendon Structure*	AFT	208	0.18 ± 0.03	0.18 to 0.17	0.028	−0.8
		MFT	72	0.2 ± 0.17	0.23 to 0.18		
	*pain*	No_Pain	112	0.19 ± 0.15	0.22 to 0.17	0.735	−0.32
		Pain	110	0.2 ± 0.16	0.23 to 0.17		
	*Sex*	Female	152	0.19 ± 0.14	0.21 to 0.17	0.389 *	−0.8
		Male	128	0.21 ± 0.16	0.23 to 0.18		
	*Jump*	CMJ	140	0.19 ± 0.03	0.2 to 0.19	0.684	−0.37
		TJT	140	0.2 ± 0.21	0.24 to 0.17		
Load_Rate/Knee	*Tendon Structure*	AFT	208	13.72 ± 9.28	15.90 to 11. 54	0.993	0.22
		MFT	72	13.72 ± 10.10	15.10 to 12.34		
	*pain*	No_Pain	112	13.70 ± 9.79	15.18 to 12.23	0.976	−0.65
		Pain	110	13.74 ± 10.06	15.65 to 11.84		
	*Sex*	Female	152	13.26 ± 9.21	14.74 to 11.79	0.412 *	−0.06
		Male	128	14.26 ± 10.63	16.12 to 12.40		
	*Jump*	CMJ	140	7.42 ± 2.23	7.79 to 7.05	0.000 *	−0.62
		TJT	140	20.02 ± 10.53	21.78 to 18.26		
Area/Knee	*Tendon Structure*	AFT	208	332.72 ± 62.2	347.34 to 318.11	0.733	−0.62
		MFT	72	336.59 ± 125.9	353.8 to 319.38		
	*pain*	No_Pain	112	336.57 ± 117.4	354.34 to 318.79	0.856	0.32
		Pain	110	334.09 ± 106.08	to 314.04		
	*Sex*	Female	152	290.42 ± 85.21	304.07 to 276.76	0.000 *	−0.73
		Male	128	389.24 ± 118.43	409.95 to 368.53		
	*Jump*	CMJ	140	313.3 ± 56.16	322.69 to 303.92	0.000 *	−0.86
		TJT	140	357.88 ± 146.39	382.34 to 333.42		

* = *p* < 0.005; CI = coeficient interval; AFT = aligned fibrillar tendon; MFT = misaligned fibrillar tendon; CMJ = counter movement jump; TJT = tuck jump test; Sec = seconds; N = Newtons.

## Data Availability

Data are contained within the article.

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
