# Peer review of "Presence of Pain Shows Greater Effect than Tendon Structural Alignment During Landing Dynamics"

_jfmk, 2025, doi:10.3390/jfmk10010074_

Round 1

Reviewer 1 Report

Comments and Suggestions for Authors

The manuscript entitled “Presence of Pain Shows Greater Effect than Tendon Structural Alignment during Landing Dynamics” was evaluated. The manuscript presents relevant information on the topic; however, some adjustments need to be made in order for it to be worthy of publication.

Below are some questions and suggestions for adjustments that should be observed:

Abstract

1- In the abstract methods, state which statistical tests will be performed in the study and present the “p” value.

2- In line 22, what does the term “VISA-P” mean? Before presenting abbreviations, please write it out in full.

3- In line 26, the name is repeated in full and the abbreviation “Joint angle (JA)”, however, this information is already in line 24, so only present the acronym.

4- Separate the words “jump.Results:” in line 28.

5- What does the acronym “ES” that appears in lines 29, 30, 33 and 34 mean? If it is a result of a statistical test, please mention which test was used.

6- Try to adjust the conclusion of the abstract to respond more directly to the objectives proposed in the study. In addition, the division of the groups needs to be clearer in the methodology, as it is not possible to clearly understand the results.

Introduction

7- The introduction presented an objective that is not the same as the abstract. Please standardize it so that both objectives are the same.

8- The sentence presented at the end of the introduction (lines 116 to 118) could be thought of and included in the conclusion of the manuscript. My suggestion is that it be removed from that location.

9- The introduction is a bit long. My suggestion is that it be condensed, leaving only the essential information for understanding the study.

Methods

10- Since this is a clinical trial, it is necessary to present in the manuscript where the study was registered and the website link. Please add this information in line 129, after the information about the ethics committee.

11- It is also important to present a flowchart of the study, starting from the selection of participants to how many are left to perform the statistical analyses.

12- In line 21 (abstract) it is mentioned that the participants were between 18 and 35 years old, however, in line 131 it is mentioned that they were between 18 and 30 years old. Please check this information and adjust it.

13- The text located between lines 153 and 196 is written in a single, very long paragraph. My suggestion is that it be subdivided into paragraphs to make it easier to read. Another suggestion would be to create subtopics for each evaluation or test mentioned. The text also did not include information related to anthropometric assessment, as well as the names of the equipment and manufacturers.

Results

14- In line 21 (abstract) it is mentioned that 14 women and 25 men participated in the study, totaling 39. In line 224 of the results, 31 are mentioned and in table 1 it says that the AFT group had 21 women and 28 men and the MFT group had 7 women and 12 men. Please check this information and adjust to the correct number of participants.

15- Adjust all tables in the article, as they are presented as a chart. In tables, they should only contain the top and bottom lines and should not be closed on the sides or have internal divisions. Another issue is the fact that the title and caption of the tables should be presented as text and not together with the table. In addition, it was verified in tables 2 and 3 that the title appears both linked to the table and in the text. 

16- In Table 1, it is necessary to present the information for men and women separately, especially those related to body mass and height, which are biologically distinct variables between the sexes. Also present the values of the body mass index so that it is possible to know a little about body composition.

17- Figure 1 is a little distorted and has low resolution, in addition, the information written in gray is very clear, please adjust it. It is also necessary to present in the figure caption which statistical analysis was performed, the “p” value and explain the acronyms presented in the figure.

Discussion

18- In lines 324, 337 and 350 of the introduction, please check whether the author citation format “Scattone et al. 2016” is correct according to the journal’s standards.

19- In clinical limitations, only leave the abbreviation “CMJ” and “TJT” in lines 382 and 383.

Author Response

Thank you for your valuable feedback on our manuscript. Please find the following answers: 

Abstract

  • In the abstract methods, state which statistical tests will be performed in the study and present the “p” value.

Thank you for the comment, the following test has been added. "A general linear model was used to evaluate joint angle (JA) differences between tendon alignment, pain and sex variables. Sample t-tests compared peak_vGRF, load time, load rate, and area based on tendon alignment, pain presence, sex, and jump. The significance of p-value is > 0.05."

  • In line 22, what does the term “VISA-P” mean? Before presenting abbreviations, please write it out in full.

Thank you for the appreciation, corrections have been made

  • 3- In line 26, the name is repeated in full and the abbreviation “Joint angle (JA)”, however, this information is already in line 24, so only present the acronym.

Thank you for the appreciation, corrections have been made

  • Separate the words “jump. Results:” in line 28.

Thank you for the appreciation, corrections have been made

  • What does the acronym “ES” that appears in lines 29, 30, 33 and 34 mean? If it is a result of a statistical test, please mention which test was used.

Thank you for th appreciation, ES has beend defined and statisiciall test mentioned

  • Try to adjust the conclusion of the abstract to respond more directly to the objectives proposed in the study. In addition, the division of the groups needs to be clearer in the methodology, as it is not possible to clearly understand the results.

The following text has been added "patellar tendon ultrasound characterization tissue (UTC) scans, in order to identify groups with misaligned tendon fibres (MTF) or aligned tendon fibres (ATF),.... as well as conclusion have been adjusted to "d Pain is a critical factor influencing greater JA during landing, particularly at the trunk, hip, and ankle joints in CMJ."...

Introduction

  • The introduction presented an objective that is not the same as the abstract. Please standardize it so that both objectives are the same.

Thank you for the appreciation, objectives have been standarized in abstract and introduction

  • The sentence presented at the end of the introduction (lines 116 to 118) could be thought of and included in the conclusion of the manuscript. My suggestion is that it be removed from that location.

Thank you for the comment; after discussing with the co-authors we would like to leave the sentence as provides information about the potential contribution of the study within the field

  • The introduction is a bit long. My suggestion is that it be condensed, leaving only the essential information for understanding the study.

The reviewer is right, although we believe that it is important to provide information about each of the sections of the introduction as they provide evidence of all the different aspects of the study: pathology, tendon structure and landing biomechanics. 

Methods

  • Since this is a clinical trial, it is necessary to present in the manuscript where the study was registered and the website link. Please add this information in line 129, after the information about the ethics committee.

Thank you for the comment, as this is an observational study and there is no intervention from the investigators or any kind of treatment, the study's registration is not mandatory. But the reviewer is right and we are registering for the study. Right now we cannot provide the registration number yet as the deadline to provide a reviewer's answer. In case the study is accepted, by the time all procedures are finalized we will be able to provide the registration number, 

  • It is also important to present a flowchart of the study, starting from the selection of participants to how many are left to perform the statistical analyses

Flow chart included in the manuscript

  • In line 21 (abstract) it is mentioned that the participants were between 18 and 35 years old, however, in line 131 it is mentioned that they were between 18 and 30 years old. Please check this information and adjust it.

Thank you for the appreciation, corrections have been made

  • The text located between lines 153 and 196 is written in a single, very long paragraph. My suggestion is that it be subdivided into paragraphs to make it easier to read. Another suggestion would be to create subtopics for each evaluation or test mentioned. The text also did not include information related to anthropometric assessment, as well as the names of the equipment and manufacturers.

Subsections have been made. There is a paragraph for each of the data collection steps. Equipment and manufacturers have been highlighted in the text.

Results

  • In line 21 (abstract) it is mentioned that 14 women and 25 men participated in the study, totaling 39. In line 224 of the results, 31 are mentioned and in table 1 it says that the AFT group had 21 women and 28 men and the MFT group had 7 women and 12 men. Please check this information and adjust to the correct number of participants.

Thank you for the appreciation, corrections have been made

  • Adjust all tables in the article, as they are presented as a chart. In tables, they should only contain the top and bottom lines and should not be closed on the sides or have internal divisions. Another issue is the fact that the title and caption of the tables should be presented as text and not together with the table. In addition, it was verified in tables 2 and 3 that the title appears both linked to the table and in the text. 

Thank you for the comment.  Tables are presented as editable tables now, instead of charts, in order to adjust editions. Data is presented with internal divisions. The title has been removed from the table.

  • In Table 1, it is necessary to present the information for men and women separately, especially those related to body mass and height, which are biologically distinct variables between the sexes. Also present the values of the body mass index so that it is possible to know a little about body composition.

We decided to present together as there were no significant changes to height and weight. 

  • Figure 1 is a little distorted and has low resolution, in addition, the information written in gray is very clear, please adjust it. It is also necessary to present in the figure caption which statistical analysis was performed, the “p” value and explain the acronyms presented in the figure.

Discussion

  • In lines 324, 337 and 350 of the introduction, please check whether the author citation format “Scattone et al. 2016” is correct according to the journal’s standards.

Thank you for the appreciation, corrections have been made

  • In clinical limitations, only leave the abbreviation “CMJ” and “TJT” in lines 382 and 383.

We have not been able to find the comments accordind to the lines. In lines 382-3 there are no mentione of CMJ or TJT

Reviewer 2 Report

Comments and Suggestions for Authors

INTRODUCTION

- The approach of the study is sufficiently justified.

- The authors have used an adequate number of references to present the theoretical framework.

MATERIALS AND METHODS

- Line 121: Of the 39 volleyball players, what was the distribution of women and men?

- Lines 122-124: where the recruitment of participants took place is indicated, but the recruitment method or procedure is not detailed.

- The text of subsection 2.2 (lines 131-135) does not refer so much to the Participants as its title indicates, but is actually the inclusion and exclusion criteria. This has to be mentioned explicitly.

- The information shown in subsection 2.4 is too compacted, dense and confusing. Information about signing the informed consent (lines 153-154) should appear before. And the three data collection steps (lines 154-158, 158-177 and 177-196) should be in separate subsections. Please also use paragraph separation (the text between lines 153 and 196 is currently in a single paragraph).

- Line 199: For this procedure, is self-citation absolutely necessary? Couldn't another reference be used?

- Line 213: replace "p 0.005" with "p < 0.005", which is how it appears in the rest of the manuscript.

RESULTS

- Line 121 vs. 224: why 8 less players? What happened to them?

- The data between line 226 and Table 1 regarding the number of subjects seems to be different. On line 228, n=38 for AFT and n=24 for MFT. However, in Table 1, n= 49 (21+28) for AFT and n=19 (7+12) for MFT.

- Units in Table 1 are missing.

- The text on lines 246-247 should be deleted; it is later repeated on lines 249-250 (its correct location being below the figure).

- Tables 2 and 3: What do the values ​​in bold mean? The units in which the results are expressed are also missing.

CONCLUSIONS

- Well presented, relying on previous limitations and practical implications.

Author Response

Thank you for your valuable feedback on our manuscript 

INTRODUCTION

The approach of the study is sufficiently justified.

The authors have used an adequate number of references to present the theoretical framework.

Thank you for the time and comments

MATERIALS AND METHODS

Line 121: Of the 39 volleyball players, what was the distribution of women and men?

Comments have been added to the text as:"professional female (n=22) and male (n=17) volleyball players.."

 Lines 122-124: where the recruitment of participants took place is indicated, but the recruitment method or procedure is not detailed.

Further detail has been added: ". Participants were recruited through verbal information from physical coaches and university staff affiliated with the Portuguese Volleyball Federation and Polytechnic Institute of Porto. Reporting "

The text of subsection 2.2 (lines 131-135) does not refer so much to the Participants as its title indicates, but is actually the inclusion and exclusion criteria. This has to be mentioned explicitly.

Tittle has been specifically mentioned 

The information shown in subsection 2.4 is too compacted, dense and confusing. Information about signing the informed consent (lines 153-154) should appear before. And the three data collection steps (lines 154-158, 158-177 and 177-196) should be in separate subsections. Please also use paragraph separation (the text between lines 153 and 196 is currently in a single paragraph).

Subsections have been made, there is a paragraph  for each of the data collection steps. information and informed consent liens have been moved above in the text

Line 199: For this procedure, is self-citation absolutely necessary? Couldn't another reference be used?

Authors decided to use a self-reference because we could not find the information to calculate the sample size. In this way facilitate replicate the methodology for future studies and systematic reviews with meta-analysis.

Line 213: replace "p  0.005" with "p < 0.005", which is how it appears in the rest of the manuscript.

Correction have been made on the text

RESULTS

- Line 121 vs. 224: why 8 less players? What happened to them?

Further detail has been added in the manuscript: "After data processing, only 31 volleyball players selected for the study due to poor tendon imaging or inaccurate kinematics recordings."

The data between line 226 and Table 1 regarding the number of subjects seems to be different. On line 228, n=38 for AFT and n=24 for MFT. However, in Table 1, n= 49 (21+28) for AFT and n=19 (7+12) for MFT.

Thank you for th appreciation, corrections have been made

Units in Table 1 are missing.

Units and * p value have been added

The text on lines 246-247 should be deleted; it is later repeated on lines 249-250 (its correct location being below the figure).

Thank you for the appreciation, corrections have been made

Tables 2 and 3: What do the values ​​in bold mean? The units in which the results are expressed are also missing.

Thank you for the appreciation, corrections have been made

CONCLUSIONS

Well presented, relying on previous limitations and practical implications.

Thank you to the reviewer for the comments

Reviewer 3 Report

Comments and Suggestions for Authors

It is an interesting topic.

The purpose of the paper is well defined.

Table 1. a) Data is presented in mean ± estandart deviation; Abreviations: VISA-P ( Victorian Institute Sports Assessment Patellar)

You wanted to write standard deviation?

It is appreciated that you presented as a limitation of the study - the ability to establish causality between landing dynamics and structural changes of the tendon. I think the small number of participants in the study should also be passed, meaning 31.

It seems, however, a slightly negligent written study, see the references.

1. (1) Theodorou, A.; Komnos, G.; Hantes, M. Patellar Tendinopathy: An Overview of Prevalence, Risk Factors, Screening, …..

 2. (2) Magnusson, S.; Langberg, H.; Kjaer, M. The Pathogenesis of Tendinopathy: Balancing the Response to Loading. Nat. Rev. …..

I appreciate the work done for this study.

Further studies are certainly needed.

My comments are only intended to make the paper better. Good luck!

Author Response

Thank you for your valuable feedback on our manuscript 

It is an interesting topic.

The purpose of the paper is well defined.

Table 1. a) Data is presented in mean ± estandart deviation; Abreviations: VISA-P ( Victorian Institute Sports Assessment Patellar)

You wanted to write standard deviation?

Thank you for the appreciation, corrections have been made

It is appreciated that you presented as a limitation of the study - the ability to establish causality between landing dynamics and structural changes of the tendon. I think the small number of participants in the study should also be passed, meaning 31.

Comments have been added to the text as: "This study's cross-sectional design and the small number of participants in the MFT group limits the ability to establish causality between landing dynamics and tendon structural changes."

It seems, however, a slightly negligent written study, see the references.

  1. (1)Theodorou, A.; Komnos, G.; Hantes, M. Patellar Tendinopathy: An Overview of Prevalence, Risk Factors, Screening, …..
  2. (2) Magnusson, S.; Langberg, H.; Kjaer, M. The Pathogenesis of Tendinopathy: Balancing the Response to Loading. Nat. Rev. …..

Thank you for the appreciation, corrections have been made

I appreciate the work done for this study.

Thank you to the reviewer for the time

Further studies are certainly needed.

My comments are only intended to make the paper better. Good luck!

We appreciate the time and comments

Round 2

Reviewer 1 Report

Comments and Suggestions for Authors

Dear authors and editor,

After analyzing the adjustments made by the authors, my suggestion is that the article be accepted for publication. However, I believe that the flowchart should include a caption explaining the acronyms and the dynamics of the study and that for publication the authors should include the clinical trial record in the work. It is also necessary to remove the grid lines from the tables.

Author Response

Thank for to reviewer 1 your time and contribution

Reviewer 1: After analysing the adjustments made by the authors, my suggestion is that the article be accepted for publication. However, I believe that the flowchart should include a caption explaining the acronyms and the dynamics of the study and that for publication the authors should include the clinical trial record in the work. It is also necessary to remove the grid lines from the tables.

Acronyms have been added to the flow chart . Registration is in process. Grid line removed. We have registered the study in clinicaltrials.gov number _ NCT06829056

Kind regards
